# Low Psychological Resilience in Older Individuals: An Association with Increased Inflammation, Oxidative Stress and the Presence of Chronic Medical Conditions

**DOI:** 10.3390/ijms22168970

**Published:** 2021-08-20

**Authors:** Ljiljana Trtica Majnarić, Zvonimir Bosnić, Silva Guljaš, Domagoj Vučić, Tomislav Kurevija, Mile Volarić, Ivo Martinović, Thomas Wittlinger

**Affiliations:** 1Department of Internal Medicine, Family Medicine and the History of Medicine, Faculty of Medicine, Josip Juraj Strossmayer University of Osijek, Huttlerova 4, 31000 Osijek, Croatia; ljiljana.majnaric@mefos.hr; 2Department of Public Health, Faculty of Dental Medicine and Health, University Josip Juraj Strossmayer, Crkvena 21, 31000 Osijek, Croatia; 3Faculty of Medicine, Josip Juraj Strossmayer University of Osijek, Huttlerova 4, 31000 Osijek, Croatia; zbosnic191@gmail.com (Z.B.); silva.guljas@gmail.com (S.G.); domagojvucicmedri@gmail.com (D.V.); tkurevija6@gmail.com (T.K.); mvolaric@gmail.com (M.V.); 4Department of Cardiology, General Hospital “Dr. Josip Benčević”, 35105 Slavonski Brod, Croatia; 5Department of Cardiothoracic Surgery, University Hospital Marburg, 35033 Marburg, Germany; ivo.martinovic@med.uni-marburg.de; 6Department of Cardiology, Asklepios Hospital, 38642 Goslar, Germany

**Keywords:** psychological resilience, inflammation, oxidative stress, aging, chronic diseases

## Abstract

The term resilience, which has been present in science for almost half a century, stands for the capacity of some system needed to overcome an amount of disturbance from the environment in order to avoid a change to another stable state. In medicine, the concept of resilience means the ability to deal with daily stress and disturbance to our homeostasis with the intention of protecting it from disturbance. With aging, the organism becomes more sensitive to environmental impacts and more susceptible to changes. Mental disturbances and a decline in psychological resilience in older people are potentiated with many social and environmental factors along with a subjective perception of decreasing health. Distinct from findings in younger age groups, mental and physical medical conditions in older people are closely associated with each other, sharing common mechanisms and potentiating each other’s development. Increased inflammation and oxidative stress have been recognized as the main driving mechanisms in the development of aging diseases. This paper aims to reveal, through a translational approach, physiological and molecular mechanisms of emotional distress and low psychological resilience in older individuals as driving mechanisms for the accelerated development of chronic aging diseases, and to systematize the available information sources on strategies for mitigation of low resilience in order to prevent chronic diseases.

## 1. Introduction—Definition of the Term Resilience and in Particular Resilience in Psychology

The term resilience has been used in various contexts and across a range of disciplines, including engineering, ecology, economics, life sciences, psychology and psychiatry, to describe the plasticity and adaptability of complex dynamic systems to adverse circumstances [1]. In animal and human physiology, this term is used to improve our understanding of neuro-endocrine and other biological responses to stressful situations, corresponding with the ability of live organisms to maintain or regain physiological homeostasis after acute stress stimuli [2]. Many experimental and animal models have been employed to date to help clarify the stress-related reactions of cells, tissues and physiological systems. Yet these models are not capable of coupling with truly human experiences by means of reconciling complex situations, where environmental stressful stimuli are modified by interactions with protective factors and individuals’ emotional and cognitive processing of these real situations [3]. The psychological concept of resilience has evolved in the late 20th century from child and developmental psychology, where it focused on personal traits that may distinguish children who are resistant to disadvantages in upbringing from those showing poor outcomes [4]. This early concept of resilience, considered as a positive personality trait, has gained further understanding from adult research, the focus of which was on persons who are resistant to post-traumatic stress disorder (PTSD) [5]. 

The results of this early resilience research demonstrate that the psychological concept of resilience has been significantly revolutionized, showing that psychological resilience is a complex construct and that there are multiple factors interacting with each other and over the lifespan which may influence outcomes in the period of time that comes after adversity [3]. Some of these identified factors include intellectual functions, events during development, traumatic events and all events across lifetime in general, social circumstances such as poverty, the adequacy of school programs, surviving natural disasters or war, cultural factors, religion and individual physical illnesses; they also include a supportive role of family and the community [3,4]. In this regard, it is known that environmental factors affect brain development during childhood, which manifests in neuronal networks, brain size and various receptor sensitivity. Favorable environmental factors protect and nurture during the sensitive childhood period, increasing resilience and reducing the hypothalamic-pituitary-adrenal (HPA) response to stress; inversely, traumatic events in childhood produce alterations in the HPA axis and increase the likelihood of mood and anxiety disorders [6,7,8]. It is necessary to mention genetics, which plays a modifying role and has additive effects by influencing individual personal characteristics, such as cognitive and emotional adaptability, wherein a tendency toward positive thinking increases resilience [9]. 

Intensive resilience research in past decades was essential to provide a broader understanding of the term resilience, which is defined as an outcome, as a dynamic process of adaptation, and as a personal trait [10,11]. Psychological resilience is increasingly seen as a dynamic process by which people do not only bounce back from adversity and avoid the development of psychopathology but ideally grow over time through experiences. To understand this process and the conditions under which positive functioning and adaptive behaviors develop, there is a need to understand the dynamic interplay and changes over the lifetime of a range of adaptive capabilities, including protective factors and mechanisms, risks, social relationships and cultural contexts, in addition to personal attributes and resources (Table 1). Recent technology advances in genetics, epigenetics and neurological imaging techniques have allowed for unprecedented opportunities in obtaining insights into the molecular biology level of resilience processes [10,11].

Current trends in resilience research focus on cross-cultural studies and marginalized social groups in order to improve understanding of the broad social context of resilience and to identify adaptive resources that may remain hidden under inhospitable social conditions [12]. In the modern world, where many communities are exposed to the stress of war and protracted conflicts, and where existing socioeconomic inequalities are being translated into health disparities, searching for alternative resilience approaches in marginalized groups might be of the utmost importance for alleviating rapidly increasing trends in health disparities. 

Finally, alongside the current understanding of resilience as a multifaceted construct, efforts are concentrated on developing models that would have the capacity to present different dimensions of resilience (psychological, social, spiritual, and biological) in an integrative and system perspective manner, while also reflecting the temporal nature of resilience processes [13].

Although resilience is a complex construct, it nevertheless can be measured, either subjectively, by means of self-assessment, or objectively, by being reported by an external observer. Additionally, a combination of both approaches can be applied [14]. Currently, there are many measurement tools, of which the most widely used is the Connor–Davidson Resilience Scale, which focuses on individual characteristics that build one’s resilience, such as persistence, durability, self-control under pressure, spirituality, flexibility and focus on goals [14,15]. The existing resilience scales are based on the assumption that the critical point in evaluating resilience is to assess the individual’s level of resilience at the time an individual is experiencing an adverse situation, either following an acute event or by overcoming chronic challenges [14]. 

## 2. Neurobiology of Stress and Resilience

The well-known description of the physiological response to acute stress, as pioneered by the physiologist Seyle a half of century ago, is based on activation of the HPA axis and its regulation by the negative feed-back loop. This description has evolved to encapsulate a more comprehensive picture of the neurological mechanisms underlying the stress-related response. These mechanisms include structural plasticity and functional remodeling of the cortical-limbic brain regions that are known to mediate higher order cognition and emotion process regulation, including the pre-frontal cortex, the ventral striatum, the amygdala, and the hippocampus, a brain region involved in stress-related information and emotion processing [2,16].

An appropriate response to stress involves the coordinated activity of the autonomic nervous system, the HPA axis, and the neural circuits in the aforementioned brain regions [2,16]. A variety of soluble mediators play a role in mediating these activities. These include: glucocorticoids; excitatory amino acids, in particular glutamate; and anxious- and reward-related neurotransmitters, such as dopamine and serotonin, as well as a variety of other regulatory molecules [2,16]. Proinflammatory cytokines, generated in the periphery under stress conditions, traffic back to the central nervous system (CNS) and promote a long-lasting inflammatory condition corresponding with anxious or depression-like behavior [16,17]. It is important to mention that these same mediators play a regulatory role and can promote pathophysiology changes in the case of persistence of the activation state or imbalanced activation [16]. It is also increasingly clear that resilience involves active biological processes and unique adaptation mechanisms that promote resilient behavior, and that resilience is not only passive reversal of pathological mechanisms [18].

The brain is the central organ which perceives what is potentially threatening for the body’s integrity, and therefore what is stressful; the brain also initiates and orchestrates behavioral and physiological responses to help an individual adapt to a stressful situation [16,17]. In this response, central and peripheral mechanisms collaborate through complex crosstalk, and a variety of neuroendocrine, metabolic, immune and inflammatory stimuli generated in the periphery can reach the CNS via afferent nerves of the autonomic nervous system or the vascular blood-brain barrier and modulate activities and the structural changes of particular CNS regions.

Adaptation to stressful situations may be successful by means of achievement of emotional and mental stability, or even mental health promotion and personal growth. This type of adaptation to stress is called “allostasis” and usually happens when stressful stimuli are mild to moderate and of a contemporary duration. Other factors promoting allostasis include the presence of a healthy brain architecture that is able to adapt to a new situation following changes in genes expression and the presence of resilient characteristics within a person, such as self-esteem and good impulse control [19]. It may be enough, e.g., for a person who is under stress to actively engage in a positive behavior such as physical activity to be able to bounce back after a short period of distress (a feeling of discomfort and loss of control after a stressful event). On the contrary, when there are multiple or chronic stressors, in particular in persons who already have a limited ability to constructively cope with adverse situations because his/her brain has been exposed to inhospitable environmental conditions in early life and might have acquired a maladaptive plasticity, this can cause “allostatic overload”, leading to enhanced wear-and-tear of the body and the brain [19]. 

It is important to mention here that allostatic processes are also sustained by epigenetic influences, which are likely to reflect global living conditions in a particular environment and enable a person to adapt to his/her environment [19]. This can mean that behaviors that are likely to be health-preserving at one point of time in life may become deleterious in later life [12]. Conditions under which this trade-off between mental and physical health occur are insufficiently clear. Living circumstances, cultural factors, and rate of social networking are all likely to have a role. This is illustrated, for example, in disparities in health between Black Americans and White Americans of European descent. Black Americans, who, over the life course, are exposed to inequalities in employment, income, and education opportunities, were found to suffer lower rates of major mental disorders, disproportionate to the higher levels of psychological distress they suffer; on the contrary, this racial group exhibits higher rates of chronic diseases in middle age and later life, and have, accordingly, lower average life expectancy [12]. Coping strategies that this racial group may practice to reduce stress-related anxiety and tension, such as smoking, drinking alcohol, or use of psychotropic drugs, together with eating foods high in fat and carbohydrates, which is usually a characteristic of poverty, may impose a cost on physical health in later life by developing pathophysiological changes associated with an unhealthy lifestyle.

## 3. Psychological Resilience in Older Individuals

A traditional view in Western culture on aging is that older age is associated with disability, frailty, and an overall decline in physical, mental and social functioning [20]. This view has begun to change in recent years, as resilience research has demonstrated that many old individuals are capable of maintaining psychological stability and well-being despite experiencing poorer financial situations, adverse events, loss of loved ones, and the burden of chronic diseases [21]. Studies show that even very old individuals (85 and older) may have high levels of resilience by means of good self-efficacy, problem solving abilities, and maintaining personal control; resilience in this age group can be even higher than in those of younger age [20,21]. Although studies show that levels of resilience in older individuals may vary depending on characteristics of the examined populations and the scale used to measure it, the general impression is that higher levels of resilience are associated with increasing age [21]. Accordingly, older people have lower rates of psychopathology than the general population [22]. High levels of resilience in older age is a phenomenon that is close to the concept some authors refer to as “the paradox of subjective well-being”, which states that levels of psychological well-being remain stable across old age despite many losses associated with aging [23]. These characteristics of older age are assumed to be due to the fact that older age is associated with gains that are based on wisdom and learning through experience, the expression of which are resilient traits such as self-confidence, autonomy in decision-making, and life management skills [24]. Affective reactions in older individuals are in general less intense than in those who are younger, reflecting their learned ability for emotion regulation [25]. Older individuals report feeling content and having a purpose in life. Their perception of stress or threatening situations is substantially different from that in younger age groups and includes fear of a sudden health decline and the loss of independence or personal control [24]. 

## 4. Psychological, Social and Biological Aspects of Resilience in Older Age and Their Associations within Successful Aging (the Biopsychosocial Model of Successful Aging)

Resilience research in older age is important from the practical perspective, as this research has revolutionized our understanding of the aging process. An early model of successful aging was simply defined as one’s being free from chronic diseases and in good physical fitness, rather than including psychological factors and subjective perception of good health [20]. However, this ideal situation of being free of chronic conditions is an exception rather than a rule in advanced age, and it has been realized that models of successful aging that would allow older individuals to express their own experiences of adaptive processes would be more realistic. We know today that being resilient does not simply mean the absence of disease. There are many models of successful aging, each tending to one aspect of successful aging, and with some models using some aspects of resilience as outcome measures [20]. Since no one of these models can comprehensively describe what means successful aging, some authors have proposed models that would be able to express differences among older individuals in attaining resilience, ultimately linking together the concept of successful aging with the concept of resilience.

Most research in gerontology today claims that conceptualization of the model of successful aging should focus on the interplay between psychological, social, and biological aspects of high resilience and the pathways that connect them (the biopsychosocial model of successful aging) (Table 2) [21,26]. The conceptualization of this model was derived from a large body of evidence suggesting that high psychological resilience and its different dimensions, including self-efficacy, positive emotions, and good social relationships, are associated with a reduced risk of chronic diseases and various other positive outcomes, such as lower decline in physical function, better mental health, lower depression, faster recovery from cardiovascular (CV) incidents, increased longevity, and lower mortality risk [21,26]. This model is expected to improve our understanding of the impact of psychological resilience (together with social resources) on the promotion of physical health (and vice versa). Understanding factors that can foster psychological resilience would inform interventions that are likely to reduce the risk of age-related chronic conditions.

In “the society-to-cell” model of resilience in older adults, the authors proposed a dynamic model of aging where multiple factors, including the intrinsic capacity of an individual as well as past and current environmental conditions, interact to create three distinct conceptualizations of resilience. These are known from the literature as resistance to a challenge, recovery from a challenge, and rebound to a challenge [27]. 

In view of these new findings, successful (or healthy) aging tends to be considered from a functional rather than disease-based perspective [28]. Functional ability is defined with regards to health-related attributes that enable older people to do what they themselves value as important for their life and is determined by the intrinsic capacity of a person, environmental (contextual) factors, and their interaction. Several measurable aspects of physical and mental capabilities of individuals have been identified to compose the intrinsic capacity index, including mobility, general vitality, cognitive function, mental health, and sensory organ functions [28]. 

Mounting evidence suggests that in older individuals, mental disorders are closely associated with physical comorbidities as an expression of lower resilience. With increasing age, the number and complexity of chronic comorbid conditions are known to increase [29,30]. With an increase in comorbidity levels, the prevalence of mental disorders also increases [31,32,33,34]. Depression is prevalently found in comorbidity patterns with some chronic diseases, such as diabetes, cardiovascular disease (CVD), cancer, rheumatoid arthritis and multiple sclerosis [35]. Notably, an association between CVD and depression is strong and seems to be mutually causal. For instance, depression is an important risk factor for CVD; when depression is in comorbidity with CVD, this increases the risk of mortality in CVD patients. Inversely, patients with CVD are more likely to develop depression than those in the general population [36]. If viewed through the lens of these facts, the presence of psychopathology in association with higher levels of comorbidity in older individuals can be considered as an expression of lower resilience, maladaptive coping, and unsuccessful aging. In this regard, in a meta-analysis where authors explored associations between high resilience and mental health, high resilience was found to be positively associated with good mental health, positive affect, and life satisfaction, and negatively associated with depression, anxiety, and pessimistic affect [37]. 

## 5. The Intersection between Psychological and Biological Resilience in Older Individuals—Significance for Successful Aging

The combined psychological and physical characteristics of resilience are considered to ultimately impact the aging process [21,26]. Differences among older individuals in the ability to attain these two aspects of resilience is a key factor in determining the interindividual variations in health and differences in health trajectories. Physical resilience, expressed as preserved mobility and good physical function, is an important element in attaining high psychological resilience, as it positively influences sense of self-coherence and self-efficacy, and boosts optimism and feelings of satisfaction with life [38]. Conversely, high psychological resilience, especially when supported by favorable environmental factors and good social relationships, can modulate an individual’s perception of stressful situations, and, accordingly, lessen physiological responses to stressful challenges. Another route by which high psychological resilience can positively affect physical resilience is by promoting health-preserving behaviors (problem-solving, physical activity, healthy diet, uptake of screening tests, sleeping well) (Figure 1) [24]. A better understanding of mechanisms operating at the intersection between the psychological and physical aspects of resilience in older individuals would increase our understanding of the nature of aging and of processes underlying the development of age-related diseases and would inform interventions aimed at attenuating the age-related decline in resilience. 

The biological processes underlying psychological resilience include adaptive mechanisms in the CNS, the HPA axis, and the immune and metabolic systems [16,19]. In some cases, exaggerated activation of these systems by “allostatic load” can promote a cascade of pathophysiologic reactions in other organs and bodily systems, increasing the predisposition of individuals who failed to adapt to stressful challenges to the development of chronic diseases [39]. It is well-accepted, currently, that chronic psychological stress accelerates aging, operating through overlapping physiological, cellular, and molecular biology mechanisms and by accelerating the decline in homeostatic reserves. Nevertheless, the current picture of the biological effects of chronic stress on the development of chronic diseases is still limited regarding responses of the neuro-endocrine and immune system (see Section 7). Furthermore, a systematic review of the cellular and molecular mechanisms of the biological effects of chronic stress on the development of chronic health conditions is still lacking [40,41]. The exception is our improved understanding of how chronic stress promotes mental disorders, depression, PTSD, and metabolic and vascular changes, ultimately leading to the development of CVD (see Section 8). 

According to the current knowledge, biological (physical) resilience is a key manifestation of aging that contributes to an increased risk of mortality with age and eventually limits the human lifespan [42]. Although biological robustness (ability to resist deviations from the normal physiological state) generally declines with age, corresponding with the definite process of transition from a healthy to a frail state and with reduction in homeostatic reserves, it is assumed that biological resilience (ability to recover after deviation) can be increased, which provides the framework for anti-aging interventions [42]. At the phenomenological level, biological resilience is marked by the ability of an older individual to restore glucose levels or blood pressure or heart rate after deviations caused by a stressor, or by other abilities, such as wound healing or survival after an adverse health event. 

Although older individuals differ from each other in longevity and the expression of aging phenotypes, rates of aging are determined by several conserved genetic pathways and biochemical processes, known as the hallmark of aging (Table 3) [43]. Experimental studies of aging have shown that genes that influence aging phenotypes and longevity are also involved in resilience. The most studied pathways are those from insulin and insulin-like growth factor-1 (*IGF-1*), protein-kinases (*AKT*), *FOXO* transcription factors, complex *mTOR* (a target of rapamycin), *p16* cell cycle inhibitor, and sirtuins (*S6K*), which all jointly decide outcomes of cell responses to stress and damage by influencing cell survival, growth, DNA repair, apoptosis, cellular senescence and autophagy (Table 3) [44,45].

Emerging approaches in measuring age-related resilience include measuring how quickly and completely an individual recovers from acute stress, such as a hip fracture. This can be accomplished by modeling composite resilience measures from longitudinal human data; this is the case with the physiological dysregulation index (PD), which integrates deviations of multiple biomarkers from their baseline/normal physiological states into one estimate reflecting the loss of homeostasis in biological networks or longitudinal analysis of blood markers [46,47,48]. Recent advances include continuous measurements of individual physiological responses from wearable devices with the reconstruction of longitudinal trajectories of resilience markers [49]. 

Previous studies demonstrated that gender does not always predict resilience, although it appears to be a factor that is associated with resilience [50]. For instance, women appear to be generally more resilient than men, as they may establish and maintain social connections through volunteering and community involvement, all of which seem to support high resilience [51]. There is also a “trade-off” between male and female robustness and survival (male-to-female health-survival paradox) [52]. This is a phenomenon in which men are healthier than women but have worse survival after adverse events. This phenomenon is explained by the fact that men spend more body reserves in their youth and middle age, at the expense of a faster decline in resilience in older age. Future research on age-related resilience should focus on efforts to select and standardize a set of biological markers that might be relevant to represent an intersection between different characteristics of physiological responses to chronic stress and aging pathways. 

## 6. Adaptation to Chronic Disease as a Process of Resilience in Older Individuals

Living with a chronic disease is a challenge, and most older people already have some chronic diseases [53]. It is essential to understand how older people adjust to and cope with chronic diseases and to understand the discomfort and limitations that come with these diseases, as it makes a significant impact on disease outcome [54]. Coping with chronic diseases is especially challenging when taking into account an ongoing and a progressive course of these diseases and the fact that they rarely stand alone, but rather as two or more comorbid conditions (which is termed multimorbidity) [53]. On the other hand, recent advances in early detection and medical care for some important chronic diseases have improved prognosis; therefore, a large range of factors, both positive and negative, influence the experience of living with chronic diseases. Thus, identifying how to optimally adjust to the ever-changeable circumstances associated with chronic diseases, and to choose realistic target outcomes at different time points in the course of disease development, is becoming more and more challenging. Many factors influence how people adapt to chronic diseases, including personal factors (personality, early life experiences), social and environmental factors (social support, role playing and relations, health care services), and disease-specific factors (chronic pain, debilitating and mortality capacity of a disease). There is a growing body of literature which focuses on coping with chronic diseases. The general conclusion is that patients who are able to engage in self-care with restorative health behaviors (good compliance with medication treatment, regular physical activity, healthy diet, problem-solving coping behaviors), maintain (alone or with support) positive affect, and reduce distress and negative emotions will likely exhibit fewer symptoms, better physical functioning, and improved psychological adjustment [55]. Current research efforts are focused on creating a global model of coping with chronic diseases which incorporates different elements of adaptation process to chronic diseases [56].

## 7. Chronic Psychological Stress—Associations with Oxidative Stress, Increased Inflammation, Multiple Organ Damage, and Development of Chronic Aging Diseases

Difficulties in overcoming physiological mechanisms triggered by a stressful situation can result in a detrimental allostatic load, ultimately leading to increased susceptibility to the development of a negative stress response as well as disease development [56]. Contrary to homeostasis, allostasis is a physiological adaptation to a stressful situation [57]. In contrast to allostatic loading, there is resilience, a defense mechanism that has the role of promoting an appropriate and non-pathological response to a stressful event [56]. Systems involved in maintaining this equilibrium are the immune system, metabolic system, autonomic nervous system and the HPA axis. The HPA axis plays a key role in coordinating neuroendocrine and systemic activity in response to stress. Activation of the immune system by various stimuli leads to an increased release of proinflammatory mediators (bradykinin, histamines, leukotrienes, serotonin, prostaglandins), thus creating a local inflammatory response and increasing the synthesis and release of proinflammatory cytokines, among which the most important role is played by interleukin-1-β (IL-1β), interleukin 6 (IL-6) and tumor necrosis factor alpha (TNF-α) [58,59,60].

Increased expression and production of proinflammatory cytokines also occurs in the central nervous system in the area of stress-sensitive regions (hippocampus, amygdala, and pre-frontal cortex) [61]. Immunologically active brain tissue cells, microglia, have the ability to monitor the environment and to recognize immunological stimuli which, in turn, results in the promotion of synaptic pruning and monocyte and lymphocyte recruitment in the perivascular area (choroid plexus), the goal of which is to overcome the local inflammatory response and tissue damage [62,63]. A significant predisposition to the development of psychological disorders in chronic stress states may be the dual susceptibility of the brain to proinflammatory cytokines: those generated centrally and peripherally (bypassing the blood-brain barrier by the microglia, astrocyte, and neuron transport system) [60,64]. Two components of the neuroendocrine system, the autonomic nervous system (ANS) and the HPA axis, have a significant role in the development of stress-induced diseases, as immune cells possess glucocorticoid and adrenergic receptors [65,66]. Unlike the immune system, which has a direct effect on the brain, the neuroendocrine system primarily leads to the body’s adaptation to a stressful event, thus affecting the regulation of blood pressure and heart rate (the effect of ANS). The neuroendocrine system is also capable of increasing the synthesis and release of glucocorticoids, which has a systemic effect [16,67].

In general, during aging, there is an increase in the level of systemic inflammation (inflammaging) due to the increased production of pro-inflammatory cytokines, such as TNF-α, IL-1β, IL-6, IL-12, IL-18, and interferons type 1 (IFNs I), which is mostly a result of cell senescence and aging of the immune system [68,69]. Current knowledge on the associations between aging and the development of age-related diseases is still incomplete and the clinical evaluation of inflammaging has not yet been standardized. It is also not sufficiently understood how resilience, or one’s ability to recover from adverse events, impacts disease course. The cumulative effect of a greater degree of inflammaging, in parallel with the loss of anti-inflammation mechanisms, is considered to increase the susceptibility to, and encourage faster progression of, age-related diseases, including diabetes type 2, cardiovascular disease (CVD), dementia and cancer. The development of these diseases results in increased vulnerability of older individuals to everyday stressors and reduced functional ability and is associated with the development of frailty syndrome (reduced homeostatic reserves in multiple organs and systems) [70]. A better understanding of these processes is essential in identifying older people who are at risk of developing age-related chronic diseases. Future research should focus on investigating lifestyle behaviors and changes in health status variables as a response to psychological distress in order to facilitate efficient interventions [71] (Figure 2). 

Oxidative stress, an imbalance between free radicals and antioxidants, accompanied by inflammation, plays a major role in the origin of all diseases and pathological conditions. Additionally, some scientific theories consider oxidative stress responsible for aging, claiming that it causes a vicious circle within mitochondria in which damaged mitochondria produce increased amounts of reactive oxygen species (ROS), leading in turn to progressive augmentation in damage [72]. The mechanism of damage caused by oxidative stress is multifactorial. For instance, generating ROS greatly contributes to vascular dysfunction due to endothelial damage by impairment of endothelial nitric oxide synthase (eNOS), a key enzyme in production of the vasodilator nitric oxide (NO) [73,74]. This is associated with activation of immune cells and their recruitment to tissue, which leads to the multiple organ failure associated with aging and the development of age-related diseases and CVD in particular (Figure 3) [75,76,77]. 

In addition, increased oxidative stress can trigger autonomous cellular pathways, leading to cellular senescence, damage and apoptosis, which together with impaired clearance of cellular waste by autophagy, contributes to shaping the aging phenotypes and represents strong pro-inflammatory stimuli. Another node of the crosstalk between increased oxidative stress and inflammation involves macrophages (components of the innate immune system). Excessive production of ROS and release of oxidized mitochondrial DNA from the stressed mitochondria have been suggested to activate NLRP3 inflammasome—an intra-cellular “danger” sensing system in the innate immune system (Figure 3). Chronic activation of macrophages, due to persistent exposure to cellular waste, is an additional strong pro-inflammatory stimulus, and contributes to the aging of the immune system [78]. Mounting evidence indicates associations between chronic psychological (psychosocial) stress (and low psychological resilience) and development of chronic aging diseases, notably including metabolic disorders, atherosclerosis and CVD [78,79]. Although it has been suggested that inflammation could be a major mechanism underlying the co-existence of depression with comorbidity of chronic diseases in older individuals, the mechanistic link is not sufficiently clear (Figure 3) [80].

To summarize, the common characteristic of all chronic diseases is a “new” form of inflammation, often called meta-inflammation, which is considered a subclinical, permanent inflammation. As a result, a metabolic cascade, including cellular oxidative stress, atherosclerotic processes, and insulin resistance, occurs and slowly generates significant deterioration in the organism [81]. Finally, according to today’s prevailing theory of aging, inflammaging, a variety of stimuli in the body operating at cellular and subcellular levels contribute to low-grade inflammation as the main driver in the acceleration of aging and the development of age-related diseases [81]. Older individuals who fail to adapt to chronic psychological stress are at risk of accelerated aging due to the enhancing and synergistic effects of physiological reactions to chronic stress on the stochastic processes of “wear-and-tear”. 

The organ system that is first susceptible to increased inflammation and oxidative stress is certainly the kidneys [82]. Previous studies have investigated the influence of oxidative stress in chronic kidney disease, where it participates by damaging glomerular microcirculation and causing ischemia. In this pathological condition, oxidative stress is in a significant mutual association with inflammation factors. During chronic inflammation, stimulation of phagocytic cells, like macrophages and neutrophils, is constant, which causes excessive production of ROS while at the same time produces redox imbalance and increases inflammation. Such an environment leads to the disturbance of renal physiological resilience, causing a gradual decrease in glomerular filtration and chronic renal impairment [83]. Furthermore, accumulation of ROS plays a major role in impaired renal function through another significant pathological process, diabetic nephropathy. After a series of medical studies on rats and biopsy samples of human kidneys, scientists confirmed the association between hyperglycemia, ROS and angiotensinogen (AGT) gene expression. High glucose levels in blood directly stimulates ROS generation and subsequently activates p38 MAPK phosphorylation. ROS and activated p38 MAPK than mediate AGT gene expression in kidney proximal tubular cells but also independently contribute to vascular dysfunction in the already mentioned pathways. Finally, accumulated intrarenal AGT initiates the development of diabetic nephropathy in type 2 diabetic rats [80,84,85]. 

## 8. The Role of Chronic Psychological Stress (and Lower Psychological Resilience) in Development of Cardiovascular Disease

The direct impact of excessive neuroendocrine activity may result in the occurrence of metabolic syndrome and the development of CVD [86,87]. 

Elevated levels of stress-mediated glucocorticoids can lead to a metabolic and vascular disorder known as insulin resistance (IR) (cell and target tissue resistence to insulin-mediated glucose metabolism). Insulin resistance is present even before the onset of the first clinical symptoms of a disease (preclinical phase) [88,89]. IR is known to be associated with arterial hypertension, dyslipidemia, hyperinsulinemia, and obesity (visceral adipose tissue). The final result of chronic stress mediated by a neuroendocrine mechanism is therefore an acceleration of atherosclerotic changes in large blood vessels, and the development of CVD and cerebrovascular disease. The mechanisms of the development of these diseases are complex, but it is thought to arise from the interactions of the inflammatory response, lipid metabolism disorders, hormonal imbalances, and activation and recruitment of macrophages [90]. Elevated levels of adrenocorticotropic hormone (ACTH), glucocorticoids, and catecholamines support the development of an inflammatory response involving macrophages. Catecholamines not only lead to the development of arterial hypertension and endothelial dysfunction by stimulating α and β adrenergic receptors and activation of the renin-angiotensin-aldosterone system, but also lead to macrophage activation via surface β adrenal receptors [91,92]. Once activated, macrophages can lead to the production of proinflammatory cytokines, creating an imbalance between pro- and anti-inflammatory cytokines, and interacting with surrounding tissue cells, such as adipocytes. This all together leads to a state of chronic low-grade inflammation and increased IR [93,94]. In addition, elevated glucocorticoid levels promote gluconeogenesis and lipolysis (dyslipidemia), leading to an excessive accumulation of visceral adipose tissue, and in this way promoting IR [95] (Figure 4).

These mechanisms can be even more emphasized in older individuals, because aging *per se* is associated with an increase in systemic inflammation [69,70]. The sources of pro-inflammatory cytokines are numerous and include cell senescence and aging of the immune system as the most important ones. To conclude, the very pathophysiological mechanism of the development of stress-induced diseases is complex and is considered to be the end result of the interaction of the immune and neuroendocrine systems (Figure 4).

## 9. Strategies for Mitigation of Low Psychological Resilience as Strategies for Prevention of Chronic Age-Related Diseases

Even though there are currently several different meanings of the term “resilience”, there is one constant connection—resilience presents a defense mechanism to mitigate the negative effect of various stressors on the mental and physical health of an individual, with the aim of preserving quality of life and dignified aging [90,91]. Current research focus is on exploring strategies that can increase psychological resilience or on behavioral responses that can mitigate the negative physiological effects of chronic psychological stress. These strategies mostly include non-pharmacological methods, such as various behavioral techniques, methods of dealing with stress and anxiety, of establishing social networks and support, and engaging in physical activity, which is known to improve moods and increase physical fitness [91]. 

Additionally, psychological interventions have the potential to alter immune function, which can be relevant to different disorders where immune function is affected. The current findings are in line with previous findings supporting the clinical relevance of reversion of altered immune function following psychological interventions [95]. 

Many studies regarding coping with chronic diseases focus on identifying indicators of positive coping outcomes, including physical functioning, social functioning, and psychological adjustment [96]. Research results suggest that encouraging positive emotions through interventions such as cognitive behavioral therapy (CBT) has a significantly greater effect on well-being than just reducing negative health-related behaviors. Positive emotions generally tend to lead to positive coping outcomes, operating through both less perceived stress and active engagement in healthy behaviors [93,97]. The inclusion of physical activities, especially in individuals with chronic diseases that affect an individual’s degree of independence, has a significant impact on promoting a healthy lifestyle [98]. However, given the different aspects of disease and the socioeconomic background of each individual, personalized resilience improvement programs have the highest potential for elderly patients. Depending on the needs of an individual, these programs may include changes in the schedule of daily activities, psychological support (e.g., in the event of death of a life partner), maintaining social contacts, involvement in community activities, volunteering and other activities aimed to help others [99]. 

In recent years, the relationships between nutrition and mental health have gained considerable interest. Previous studies confirmed that older individuals with a healthy dietary pattern who consumed high intake of foods with high dietary antioxidant capacity, such as vegetables, fruits, coffee, and green tea, had a lower risk of developing frailty and less psychological distress when adjusted for other lifestyle behaviors, well-being, health status, physical functioning and social support [100,101,102]. As many researchers have confirmed the positive effects of the multidisciplinary approach in improving resilience in the elderly, in Japan, there has been a holistic community healthcare program, which is based on exposure of older individuals to the natural environment [103]. These concepts are based on the theory that our physiological functions are adapted for natural surroundings and that artificially developed urban infrastructure contributes to the development of stress. Active environmental exposure methods, such as nature and eco-therapies like forest bathing or “shinrin yoku”, which means “taking in the forest atmosphere through all of our senses”, have emerged; they can support resilience by influencing the activity of the nervous, immune and endocrine systems [104]. These approaches are expected to have an important future role in preventive healthcare.

Although these methods are promising, social support has been confirmed to have a crucial role. It is necessary to develop methods for better inclusion of the elderly in the social network, as this can reduce feelings of uncertainty and helplessness due to poorer functionality and the presence of multiple comorbidities. Finally, educating older individuals to better understand the meaning and structures of resilience may help strengthen their self-efficacy for disease management, help overcome problems in relation to health, and improve the quality of life. Additionally, attention should be paid to comprehensive, multidisciplinary, and multi-level efforts across disciplines and sectors to enhance the health and well-being trajectories of those moving into the ranks of aging society. 

## 10. Conclusions

This review presents the basis for a better and more comprehensive understanding of the phenomenon termed resilience (the ability to bounce back from adversities), and its health-related importance, specifically for an older part of the population. This paper reveals pathophysiology- and molecular-based mechanisms of emotional distress and low psychological resilience in older individuals as a driving mechanism for the accelerated development of chronic aging diseases. Although mounting evidence indicates associations of chronic psychological stress and low psychological resilience with accelerated aging and the development of chronic aging diseases, and there are pieces of evidence indicating the mediating role of increased oxidative stress and chronic inflammation in the development of chronic diseases, the clear mechanistic links and understanding of the full picture is still lacking. In particular, it is not sufficiently known to what extent there is an overlap between physiological reactions to chronic stress and mechanisms that are inherent to the aging processes. This area of research is especially challenging in older individuals because of the contradictories that exist between globally increasing psychological resilience with advancing age, relatively high rates of depression and anxiety among older patients with multiple comorbidities, and insufficient knowledge of the sources of chronic stress in older people and how these potential sources are quantified. Finally, as results of recent research have shown, living with chronic diseases may not impair longevity, functional capability and the quality of life of older individuals. Under the influence of these findings, our understanding of the concept of successful aging has begun to change. Current research efforts are focused on creating models of change that incorporate different factors associated with resilience and relate them to specific stressful challenges and different resilience outcomes. This approach could bring new insights into associations between living conditions and subjective perceptions of life with trajectories of health in an older population. This paper also synthesizes the available information sources on the strategies for mitigation of low psychological resilience in order to prevent chronic diseases. As there are currently no defined programs to improve resilience in older individuals, we support the idea of an integrated health care system infrastructure to ensure a comprehensive care program for an older part of the population. 

## Figures and Tables

**Figure 1 ijms-22-08970-f001:**
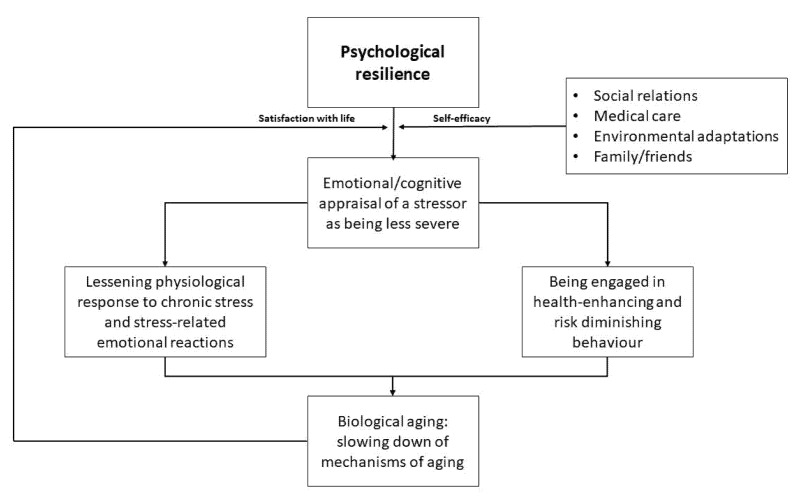
The intersection between psychological and biological resilience in older individuals.

**Figure 2 ijms-22-08970-f002:**
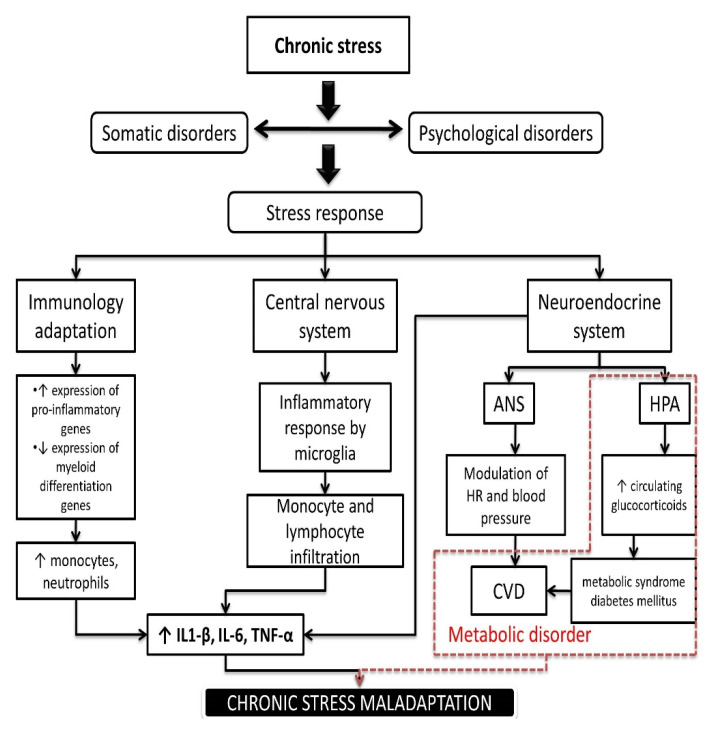
Pathophysiological complexity and mechanism of stress vulnerability. Abbreviations; IL-1β—interleukin-1-β; IL-6—interleukin 6; TNF-α- tumor necrosis factor alpha; HPA—hypothalamic-pituitary adrenal axis; ANS—autonomic nervous system; CVD—cardiovascular diseases.

**Figure 3 ijms-22-08970-f003:**
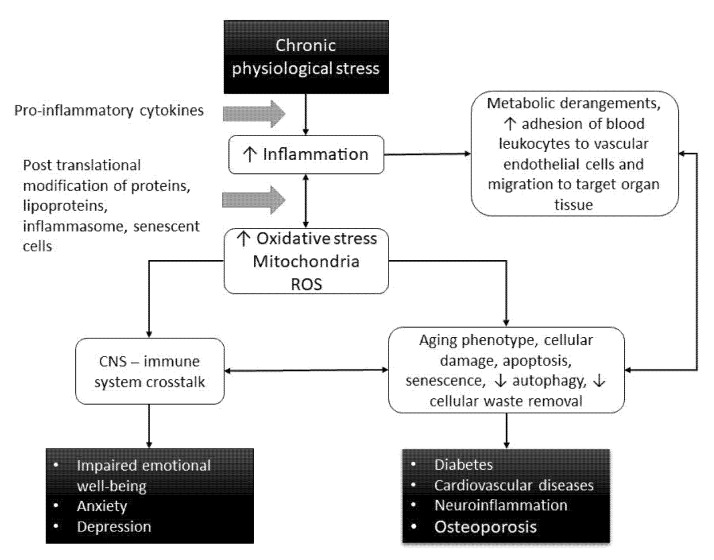
Complex understanding of the mechanism of damage caused by oxidative stress.

**Figure 4 ijms-22-08970-f004:**
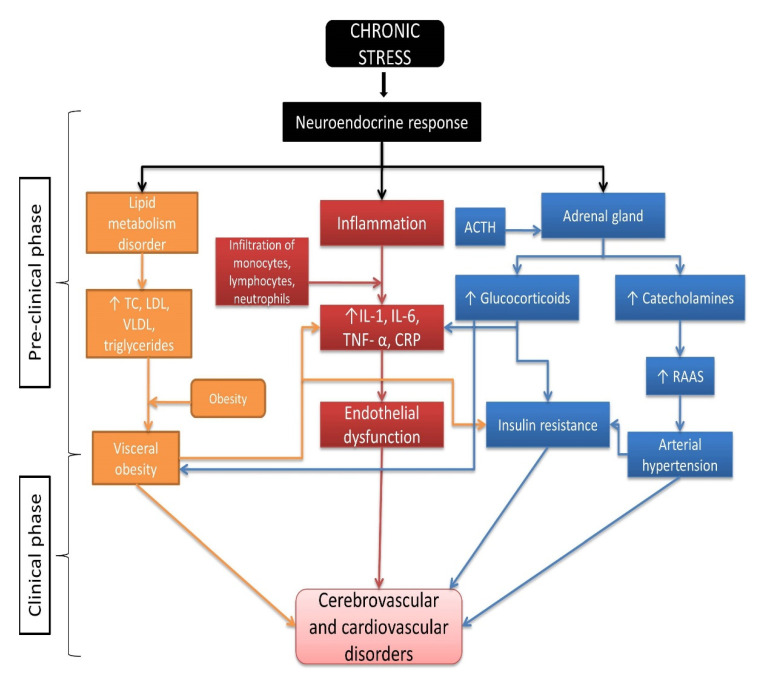
Multifactorial nature of neuroendocrine response in the onset and development of cerebrovascular and cardiovascular diseases. Abbreviations; ACTH—adrenocorticotropin; CRP—C reactive protein; IL-1—interleukin 1; IL-6—interleukin 6; LDL—low-density lipoprotein; RAAS—renin–angiotensin–aldosterone system; TC—total cholesterol; TNF-α—tumour necrosis factor α; VLDL—very-low-density lipoprotein.

**Table 1 ijms-22-08970-t001:** Factors and mechanisms of resilience across the lifespan.

Individual Level	Social Level	Environmental Level	Neurobiological Level
- Parental monitoring - Attachment - Adaptive coping (problem-solving) - Appraising a situation positively - Emotional regulation - Positive thinking - A sense of self-efficacy - Self-esteem - Optimism/hope - Hardiness - Creativity - Life management skills - Perceived social - support	- Social support - Trusting relationships - Belonging - Cultural values - Religion/spirituality - Stable living conditions - Education - Academic attainment - Financial resources - Higher socioeconomic status - Housing/transportation - Learning (life skills)	- Health services - Medical care - Healthy environment - Voices and strengths of marginalized groups	- Genetics - Epigenetics - Neural circuits - Physiological processes - (stress-related responses) - Cellular and molecular adaptation processes

**Table 2 ijms-22-08970-t002:** Characteristics of psychological, social, and physical resilience found to be associated with positive health-related outcomes.

Characteristics of High Psychological Resilience	Characteristics of High Social Resilience	Characteristics of High Physical Resilience
- Adaptive (problem-solving) coping styles - Positive emotions - Satisfaction with life - Optimism and hopefulness	- Close ties with family and friends - Community involvement - A sense of purpose (social role)	- Being mobile - Being independent in activities of daily living - A sense of being in a good health

**Table 3 ijms-22-08970-t003:** Conserved aging processes and pathways, as well as their role in shaping resilience and its decline.

Hallmarks of Aging	Aging Processes That Contribute to the Decline in Resilience	Aging Signaling Pathways That Work Together to Influence Cell Responses to Stress/Damage
- Genomic instability - Telomere shortening - Epigenetic alterations - Loss of proteostasis - Deregulated nutrient sensing - Mitochondrial dysfunction - Cellular senescence - Stem cell exhaustion - Altered intercellular communication	- Depletion of exhaustible body reserves - Slowdown of physiological processes and responses that delays recovery - Imperfect mechanisms of cell/tissue repair and cleaning (accumulation of damage and allostatic load that contributes to progressive dysregulation of body systems)	- *IGF-1/AKT/FOXO3* (nutritient sensing and signaling regulation of cell survival, growth and apoptosis, DNA repair) - *tp53/p21/p16* (senescent pathway regulating apoptosis, cellular senescence and autophagy) - *mTOR/S6K* (central regulator of longevity, regulating energy homeostasis, cellular senescence, stem cells, autophagy, cell survival and growth)

## Data Availability

No applicable.

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
