# Peer review of "Low Psychological Resilience in Older Individuals: An Association with Increased Inflammation, Oxidative Stress and the Presence of Chronic Medical Conditions"

_ijms, 2021, doi:10.3390/ijms22168970_

Round 1

Reviewer 1 Report

In the present study, the authors described how the low resilience promote aging related diseases in older people and how we improve resilience to prevent these chronic diseases. They have done a great job summarizing mechanism of chronic stress in the development of cerebrovascular and cardiovascular diseases, and the cross talk between immune and neuroendocrine systems under stress condition. The manuscript is well written. However, it would be more interesting to the readers if the authors can depict the strategies to prevent low resilience (Section 5).

Author Response

In the present study, the authors described how the low resilience promote aging related diseases in older people and how we improve resilience to prevent these chronic diseases. They have done a great job summarizing mechanism of chronic stress in the development of cerebrovascular and cardiovascular diseases, and the cross talk between immune and neuroendocrine systems under stress condition. The manuscript is well written. However, it would be more interesting to the readers if the authors can depict the strategies to prevent low resilience (Section 5).

Thank you for your suggestions.

In order to respond to your demands, we have rewrote the first five chapters again and throughly revised three chapters. Also we have added two new informative pictures and tables to more crearly present the tropic.

In section 5 we have depicted the strategies to prevent low resilience regarding nutrition, social measurements and some inventive programs like therapy with nature.

Reviewer 2 Report

The authors have attempted a review to bring together psychological resilience in older individuals with inflammation and oxidative stress. Although the concept is very novel, The authors in no way have made this correlation in this paper. The ideas are diffused throughout the paper, making it very confusing to the reader. Additionally, the molecular mechanisms described here are poor, making it not balanced for the genre of the journal.

The conclusion take a whole different tangent with claims made that are not addressed anywhere in the paper. unfortunately, the review does not present a comprehensive understanding of resilience. 

Overall, a better understanding of oxidative stress and inflammation mechanisms in conjunction with its association with resilience needs to be better represented here

Author Response

The authors have attempted a review to bring together psychological resilience in older individuals with inflammation and oxidative stress. Although the concept is very novel, The authors in no way have made this correlation in this paper. The ideas are diffused throughout the paper, making it very confusing to the reader. Additionally, the molecular mechanisms described here are poor, making it not balanced for the genre of the journal.

The conclusion take a whole different tangent with claims made that are not addressed anywhere in the paper. unfortunately, the review does not present a comprehensive understanding of resilience. 

Overall, a better understanding of oxidative stress and inflammation mechanisms in conjunction with its association with resilience needs to be better represented here

Thank you for your suggestions.

In order to respond to your demands, we have rewrote the first five chapters again and throughly revised three chapters. Also we have added two new informative pictures and tables to more crearly present the tropic.

We have made more clear correlation between psychological resilience in older individuals with inflamation and neuro-endocrine processes.

We have described molecular mechanisms in more detail and connected them more clearly with resilience.

Reviewer 3 Report

This is an excellent work on resilience in older individuals. You focus on the decisive molecular mechanisms and mediators. Oxidative stress and chronic degenerative inflammation are identified as important factors in inducing conditions, disorders and diseases that are associated with premature aging and subjective perception of decreasing health.

Your essay is a timely contribution to a better understanding and clearly demonstrates the utmost importance and relevance of the emerging research in the field.

Please add an introducing statement that reflects on the largely preliminary evidence. The neoliberal concept of resilience has been sharpely critizised and Elder for instance states in relation the experience of the great depression: "Not even great talent and industry can ensure life success over adversity without opportunity." Thus, positive thinking in a desperate environment is bullshit. Discrimination and violence towards vulnerable populations of the society can lead to traumatic experiences. Please add a short discussion of this including economic factors driving low psychological resilience in the elderly.

The major strength of your manuscript is the precise and concise discussion of the molecular mediators that drive the mechanisms of accelerated aging and the associated chronic degenerative diseases. The aspect of coping with adversity could be covered in greater detail on the molecular level with describing the biochemical and physiological markers of such adaptation in the aging population.

Psycho- and physiotherapies as well as nutrition are briefly discussed and a more detailed description on the new approaches of holistic healthcare  could improve your paper.

Overall, I am very impressed with your contribution that can guide future research in the field

Specific suggestions for minor changes

Line 235: a neuroendocrine mechanisma (missing n)

Line 236: arterosclerotic or artheriosclerotic? (r instead of n)

Line 248: here you could include a short paragraph on silent inflammation and metainflammation or permanent inflammation. Please see the work of Koch, 2020, attached, for describing this concept.

Line 293 ff: The lifestyles of health and sustainability (LOHAS) is also characterized by active prevention and an improved nutrition. A short discussion on this and the lifestyles of voluntary simplicity and downshifting could be helpful here.

Line 313: Excellent paragraph on nutrition: fully agree! Could be expanded too.

Line 344: as well as reduced depression OR associated with reduced depression?

Line 359: Japan has introduced the holistic community healthcare, a concept originally proposed by Miguel A. Pappolla and Burkhard Poeggeler. The success has been demonstrated by teh large increase in life and health span. Active environmental exposures such as nature therapies and eco therapies such as forest bathing or shinrin yoku have emerged and they can support resilience (Song et al., 2016, Physiological effects of nature therapy: a review of the research in Japan. You could refer to this already existing superior healthcare infrastructure in Japan.

Author Response

This is an excellent work on resilience in older individuals. You focus on the decisive molecular mechanisms and mediators. Oxidative stress and chronic degenerative inflammation are identified as important factors in inducing conditions, disorders and diseases that are associated with premature aging and subjective perception of decreasing health.

Your essay is a timely contribution to a better understanding and clearly demonstrates the utmost importance and relevance of the emerging research in the field.

Please add an introducing statement that reflects on the largely preliminary evidence. The neoliberal concept of resilience has been sharpely critizised and Elder for instance states in relation the experience of the great depression: "Not even great talent and industry can ensure life success over adversity without opportunity." Thus, positive thinking in a desperate environment is bullshit. Discrimination and violence towards vulnerable populations of the society can lead to traumatic experiences. Please add a short discussion of this including economic factors driving low psychological resilience in the elderly.

The major strength of your manuscript is the precise and concise discussion of the molecular mediators that drive the mechanisms of accelerated aging and the associated chronic degenerative diseases. The aspect of coping with adversity could be covered in greater detail on the molecular level with describing the biochemical and physiological markers of such adaptation in the aging population.

Psycho- and physiotherapies as well as nutrition are briefly discussed and a more detailed description on the new approaches of holistic healthcare  could improve your paper.

Overall, I am very impressed with your contribution that can guide future research in the field

Specific suggestions for minor changes

Line 235: a neuroendocrine mechanisma (missing n)

Line 236: arterosclerotic or artheriosclerotic? (r instead of n)

Line 248: here you could include a short paragraph on silent inflammation and metainflammation or permanent inflammation. Please see the work of Koch, 2020, attached, for describing this concept.

Line 293 ff: The lifestyles of health and sustainability (LOHAS) is also characterized by active prevention and an improved nutrition. A short discussion on this and the lifestyles of voluntary simplicity and downshifting could be helpful here.

Line 313: Excellent paragraph on nutrition: fully agree! Could be expanded too.

Line 344: as well as reduced depression OR associated with reduced depression?

Line 359: Japan has introduced the holistic community healthcare, a concept originally proposed by Miguel A. Pappolla and Burkhard Poeggeler. The success has been demonstrated by teh large increase in life and health span. Active environmental exposures such as nature therapies and eco therapies such as forest bathing or shinrin yoku have emerged and they can support resilience (Song et al., 2016, Physiological effects of nature therapy: a review of the research in Japan. You could refer to this already existing superior healthcare infrastructure in Japan.

 Thank you for your suggestions.

In order to respond to your demands, we have rewrote the first five chapters again and throughly revised three chapters. Also we have added two new informative pictures and tables to more crearly present the tropic.

We have included and explained socio-economic factors that have impact on low psychological resilience in the elderly.

We have made more clear correlation between psychological resilience in older individuals with inflamation and neuro-endocrine processes and described molecular mechanisms in more detail and connected them more clearly with resilience.

Holistic healthcare approach and the role of nutrition in resilience are discussed in more detail.

Regarding specific suggestions:

In line 235: we corrected the word neuroendocrine where letter n was missing

In line 236: we corrected the word arterosclerotic (r instead of n)

In line 248: we have included a paragraph on silent inflammation and we have refered on the work of Koch, 2020 for describing this concept.

In line 293 and 313 ff: We have included discussion on nutrition and resilience with more widely explanations.

In line 344: we have corrected the sentence.

In line 359: We mentioned that Japan has introduced the holistic community healthcare, and explained it in a few sentences nature therapies such as forest bathing or shinrin yoku and their role in supporting resilience (Song et al., 2016, Physiological effects of nature therapy: a review of the research in Japan.)